# Cytokine Profiling of Amniotic Fluid from Congenital Cytomegalovirus Infection

**DOI:** 10.3390/v14102145

**Published:** 2022-09-28

**Authors:** Nicolas Bourgon, Wendy Fitzgerald, Hugues Aschard, Jean-François Magny, Tiffany Guilleminot, Julien Stirnemann, Roberto Romero, Yves Ville, Leonid Margolis, Marianne Leruez-Ville

**Affiliations:** 1Service d’Obstétrique—Maternité, Chirurgie, Médecine et Imagerie Fœtales, Hôpital Necker Enfants Malades, GHU Paris Centre, AP-HP, F-75015 Paris, France; 2Équipe d’Accueil «FŒTUS» 73-28, Université Paris Cité, F-75015 Paris, France; 3Section of Intercellular Interactions, Eunice Kennedy Shriver National Institute of Child Health and Human Development, National Institutes of Health, U.S. Department of Health and Human Services, Bethesda, MD 20892, USA; 4Department of Computational Biology, Institut Pasteur, Université Paris Cité, F-75015 Paris, France; 5Program in Genetic Epidemiology and Statistical Genetics, Harvard T.H. Chan School of Public Health, Boston, MA 02115, USA; 6Service de Réanimation Néonatale, Hôpital Necker Enfants Malades, GHU Paris Centre, AP-HP, F-75015 Paris, France; 7Laboratoire de Virologie, Centre National de Reference des Herpes Virus-Laboratoire Associé Infection Congénitale à Cytomégalovirus, Hôpital Necker Enfants Malades, GHU Paris Centre, AP-HP, F-75015 Paris, France; 8Perinatology Research Branch, Division of Obstetrics and Maternal-Fetal Medicine, Eunice Kennedy Shriver National Institute of Child Health and Human Development, National Institutes of Health, U.S. Department of Health and Human Services, Bethesda, MD 20892, USA; 9Department of Obstetrics and Gynecology, University of Michigan, Ann Arbor, MI 48109, USA; 10Department of Epidemiology and Biostatistics, Michigan State University, East Lansing, MI 48824, USA; 11Center for Molecular Medicine and Genetics, Wayne State University, Detroit, MI 48202, USA; 12Detroit Medical Center, Detroit, MI 48201, USA; 13Department of Obstetrics and Gynecology, Florida International University, Miami, FL 33199, USA

**Keywords:** congenital cytomegalovirus infection, cytokines, extracellular vesicles, amniotic fluid

## Abstract

Background: Congenital cytomegalovirus (cCMV) infection is frequent and potentially severe. The immunobiology of cCMV infection is poorly understood, involving cytokines that could be carried within or on the surface of extracellular vesicles (EV). We investigated intra-amniotic cytokines, mediated or not by EV, in cCMV infection. Methods: Forty infected fetuses following early maternal primary infection and forty negative controls were included. Infected fetuses were classified according to severity at birth: asymptomatic, moderately or severely symptomatic. Following the capture of EV in amniotic fluid (AF), the concentrations of 38 cytokines were quantified. The association with infection and its severity was determined using univariate and multivariate analysis. A prediction analysis based on principal component analysis was conducted. Results: cCMV infection was nominally associated with an increase in six cytokines, mainly soluble (IP-10, IL-18, ITAC, and TRAIL). EV-associated IP-10 was also increased in cases of fetal infection. Severity of fetal infection was nominally associated with an increase in twelve cytokines, including five also associated with fetal infection. A pattern of specific increase in six proteins fitted severely symptomatic infection, including IL-18soluble, TRAILsoluble, CRPsoluble, TRAILsurface, MIGinternal, and RANTESinternal. Conclusion: Fetal infection and its severity are associated with an increase in pro-inflammatory cytokines involved in Th1 immune response.

## 1. Introduction

Congenital cytomegalovirus (cCMV) infection is the most common congenital infection, affecting 0.5–2% of live newborns worldwide [1]. cCMV infection is characterized by a variable severity, ranging from asymptomatic to severe neurological disabilities or perinatal death [2,3,4,5,6,7]. Sequelae are limited to cases following maternal infection before 14 weeks of gestation [8]. In such cases and despite changes in definitions over time, around 30% of newborns are considered symptomatic at birth, with sensorineural impairment (hearing loss (SNHL) and vestibulitis) in 10–15% and with 10–25% suffering more severe neurological damage including intellectual disability or developmental delay [5,9,10,11,12,13,14,15,16,17,18,19]. The prenatal assessment of infected fetuses is based on prenatal ultrasound, MRI, and biological data, including fetal thrombocytopenia [20,21,22,23,24,25,26,27]. Outside severe cerebral features on prenatal imaging, which are associated with a poor outcome, the prediction of neonatal status is limited, and this uncertainty weighs heavily on prenatal counselling [2]. The identification of new prognostic markers in amniotic fluid that is sampled by amniocentesis for the diagnosis of fetal infection could improve timely prenatal assessment of infected fetuses. Among biological processes involved in the innate immunity, many cytokines are involved in the immune control of cCMV infection in fetuses and immunodeficient adults [28].

The main objective of this study was to investigate all fractions of intra-amniotic cytokines in cCMV infection according to the severity at birth in order to identify suitable candidate biomarkers.

## 2. Materials and Methods

### 2.1. Study Population

All subjects were enrolled in our Fetal Medicine Unit at Necker Hospital (AP-HP hospitals of Paris and Paris-Cité University) between December 2011 and December 2017. Patients provided written informed consent, and studies were approved by the local ethics committee and registered in the clinicaltrial.gov website as NCT03090841 (BiocCMV) and NCT01651585 (CYMEVAL2). According to the study protocol, amniotic fluid that was not used for clinical management-related investigations was stored for research purposes. The women included were referred following maternal primary infection (MPI) within 2 months prior to conception or in the first trimester of pregnancy. They underwent amniocentesis to diagnose cCMV infection. For each infected fetus, a negative control with a non-infected and euploid fetus was included and matched accordingly to fetal gender and gestational age at amniocentesis. All women had no relevant medical history, especially no immune disorders or treatment affecting immunity.

### 2.2. Diagnosis of MPI, Fetal, and Neonatal Infection

The timing of MPI was determined as previously described, using an in-house algorithm based on CMV IgG and IgM antibody concentrations (LIAISON XL CMV IgG II and IgM, Diasorin, Antony, France) and IgG avidity (LIAISON CMV IgG Avidity II and/or VIDAS CMV IgG avidity II, BioMerieux, Marcy L’Etoile, France) [29].

Infected fetuses were defined by a positive CMV DNA PCR on amniotic fluid sampled by amniocentesis after 17 weeks of gestation and at least 8 weeks following maternal primary infection. At birth, neonatal infection was confirmed by CMV DNA PCR in neonatal blood, urine, and saliva. All virological tests were performed in our expert virology laboratory.

### 2.3. Clinical Classification of Infected Newborns

The pregnancy outcome was determined based upon prenatal ultrasound performed every two weeks from referral to delivery. Magnetic resonance imaging (MRI) of the fetus was performed at 28 and 32 weeks. Cordocentesis for analysis of the fetal blood was performed by ultrasound guided umbilical funipuncture at 20–28 weeks. At birth, infected fetuses were classified into two groups: symptomatic or asymptomatic newborns. Newborns with at least one abnormal neonatal feature and deceased fetuses following termination for severe brain lesions, as confirmed by postmortem examination or spontaneous intrauterine fetal death were considered symptomatic. Two sub-groups of symptomatic neonates were defined: (1) neonates with an abnormal clinical and/or complementary investigation, and (2) neonates with more severe neurological impairment as well as stillbirth or terminated fetuses with confirmed severity of brain lesions postmortem. The protocol of neonatal assessment is detailed in Appendix A.

### 2.4. Preparation of Extracellular Vesicle Fractions

Amniotic fluid samples stored in the virology laboratory of Necker University Hospital at −80 °C, were transported in a temperature-controlled device (−80 °C on dry ice) to the University of Bethesda. Preparation of extracellular vesicle was made using the methodology previously described by Bhatti et al. using Exoquick-TC™ (System Biosciences, SBI, Palo Alto, CA, USA) to sediment extracellular vesicles according to manufacturer’s instructions [30,31]. The supernatant fluid was collected into a separate tube for subsequent immunoassay later the same day (‘soluble fraction’). The pellet was centrifuged again at 1500× *g* for 5 min, and the supernatant fluid was aspirated. The pellet was resuspended in 130 μL phosphate-buffered saline (PBS, pH7.4) for subsequent analyte assay of extracellular vesicles (‘surface’ and ‘internal’ fraction). Extracellular vesicles were lysed by Triton™ X-100 at final concentration of 0.5%.

### 2.5. Cytokines Concentrations Measurement

Inflammatory mediator concentrations were determined using an in-house multiplexed bead-based assay, as previously described, with minor modifications for 38 cytokines (list available in Appendix B) [30,32]. All antibody pairs and protein standards were purchased from R&D Systems (Minneapolis, MN, USA), except those for IFN-β (Abcam, Cambridge, UK). Magnetic beads (Luminex Corporation, Austin, TX, USA) with distinct spectral signatures (regions) were coupled to analyte-specific capture antibodies according to manufacturer’s recommendations in 96-well flat bottom plates (Nunc, ThermoFisher) and incubated at 4 °C overnight. Plates were washed on a magnetic plate washer (405 TS, Biotek Winooski, VT, USA) followed by incubation with polyclonal biotinylated anti-analyte antibodies and streptavidin-phycoerythrin (ThermoFisher Scientific). Beads were resuspended in PBS and read on a Luminex 200 analyzer (Luminex Corporation) with acquisition of 100 beads for each region and analyzed using Bioplex Manager software (Bio-Rad Laboratories, Hercules, CA, USA).

Analyte concentrations (pg/mL) were determined using five parameters regression algorithms and expressed as the mean pg/mL ± S.E. Concentrations were corrected for dilution by ExoQuick-TC™ or Triton™ X-100. Extracellular vesicle luminal content (‘internal fraction’) was calculated as [analyte content of lysed vesicle] − [analyte content of intact vesicles (‘surface fraction’)].

### 2.6. Statistical Analysis

Considering sample size, we first conducted a data pre-processing, filtering out all proteins with less than 20% non-zero value. A total of 77 out of 114 (67%) proteins remained following this filtering. The distributions of all proteins, defined by their variance and proportion of non-zero values, are described in Appendix A. To provide an overview of the association profile, we then conducted a systematic univariate marginal association analysis between all available variables, including proteins, but also other measured biomarkers and clinical factors, and the two outcomes, infection and severity, using logistic and linear regressions, respectively.

Because of the large number of proteins as compared to the sample size, prediction of clinical outcomes was performed after a data dimension reduction based on principal component analysis (PCA). Note that principal components were derived based on the covariance matrix of proteins, therefore giving higher weight to proteins with higher variance. The cumulative variance explained by principal components applied to all proteins and by cellular classes are provided in Appendix A. For all sets considered, 10 PCs or less were necessary to explain 80% of the total phenotypic variance.

For infection, the prediction model was derived from multiple logistic regressions using five PCs, and prediction accuracy measured using the area under the roc curve (AUC). Given the limited sample size, the AUC was derived as the average over 50 rounds of cross-validation to limit overfitting. For each round, a training set including 80% of the data was randomly chosen to estimate regression coefficients, and the remaining 20% was used as a test set.

For severity, the prediction models were derived from multiple linear regressions, and prediction accuracy was measured as the adjusted R-squared. We investigated models including 1–40 PCs, using PCA derived in all proteins or within cellular classes. We further assessed the significance of the adjusted R-squared through permutations, where multiple regression was derived over 200 replicates after shuffling the severity values.

All analyses were performed using R. *p*-values were considered statistically significant if below the Bonferroni correction threshold for a baseline alpha value of 0.05.

## 3. Results

### 3.1. Description of the Cohort

We enrolled 80 pairs of women and fetuses/newborns in this study, including 40 infected fetuses and 40 negative controls. Clinical characteristics were similar between cases and controls (Table 1). Among infected cases, 9 (12.5%) were asymptomatic and 31 (77.5%) were symptomatic, including 13 and 18 with non-severe and severe infections. Neonatal assessment of infected fetuses is detailed in Appendix A.

### 3.2. Cytokines Profiling in Amniotic Fluid of Infected Fetuses

Concentrations of relevant cytokines and comparison between infected and noninfected fetuses are summarized in Table 2 (full in Appendix A).

Univariate analysis identified a nominally significant difference for six proteins in case of cCMV infection. Most of them were soluble in the amniotic fluid: IP-10, IL-18, ITAC, and TRAIL. In addition, we identified a nominally significant difference for the extracellular vesicle-associated IP-10, both in the internal compartment and in the surface. IP-10 further reached the very stringent Bonferroni-corrected *p*-value threshold for both infection and severity (*p* < 0.0003). For each protein previously listed, cCMV infection was associated with an increased concentration in amniotic fluid (Figure 1).

### 3.3. Cytokines Profiling in Amniotic Fluid of Symptomatic Newborns

Concentrations of relevant cytokines and comparison between symptomatic and asymptomatic newborns are summarized in Table 3 (full dataset in Appendix A).

Twelve proteins were significantly correlated to a symptomatic status at birth in the univariate analysis: four were soluble proteins (IP-10, IL-18, TRAIL, and CRP) and eight were associated with EV, mostly located within the EV (IP-10, IL-6, MCP1, MIG, and RANTES). Five cytokines were previously associated with fetal infection (IP-10_internal_, IP-10_surface_, IP-10_soluble_, IL-18_soluble_, and TRAIL_soluble_) (Figure 2). A pattern with a specific increase in cases of severe symptomatic infection was identified for six proteins (IL-18_soluble_, TRAIL_soluble_, CRP_soluble_, TRAIL_surface_, MIG_internal_ and RANTES_internal_).

Severe fetal infection was also associated with fetal thrombocytopenia (*p* < 0.001). The amniotic CMV viral load and fetal liver tests were not associated with fetal symptoms in this series.

### 3.4. Prediction Analysis 

We first assessed the prediction accuracy of fetal infection using a model based on five principal components (PCs) derived from all proteins (Figure 3a). The average AUC (area under the ROC curve) across all cross-validation equals 0.72 (SD = 0.090). In comparison, the AUC derived from a null model equals 0.50 (SD = 0.123), confirming the validity of our estimation. Using this null model, we derived a one-sided Z-score test for the observed AUC, which suggests a nominally significant prediction (*p*-value = 0.037). We then derived the prediction accuracy of various models for the severity of cCMV using 1–40 PCs derived from either intern, surface, soluble, or all proteins jointly (Figure 3b and Appendix A). The soluble-protein based and all-proteins models performed the best, explaining up to 41% of the total variance of severity. We compared these results against a null model using randomized severity status, which again, confirmed the calibration of our estimation. Together, those two analyses confirm a strong potential for using cytokines to predict both infection and severity.

## 4. Discussion

The natural history of cCMV infection is complex, and the pathophysiology of fetal injury is incompletely understood [28]. Immune response to cCMV involves both innate and adaptative immunity in the mother, placenta, and fetus at each step of the vertical transmission [33]. Among biological processes involved in the innate immunity, many cytokines appear essential to control cCMV infection as well as human CMV (hCMV) in immunodeficient adults. Only one previous study investigated cytokines profiling in amniotic fluid collected in eight infected fetuses at midtrimester and reported higher interferon gamma-induced protein 10 (IP-10) concentrations in infected cases [34].

Extracellular vesicles (EVs) include a wide spectrum of lipidic cell-derived membranous structures [35]. EVs are involved in many physiological and pathological processes, especially to mediate paracrine inter-cellular communication by carrying various types of molecules including nucleic acids, proteins, lipids, and other metabolites [35,36,37,38,39]. EVs are present in many biological fluids, including amniotic fluid [40,41], because they are released by fetal epithelial cells from skin, urine, and lungs [42,43,44,45,46,47,48]. Recent data suggested that intraamniotic EV could contain biomarkers relating to a wide spectrum of fetal disorders including bacterial intra-amniotic infection [30,40,45,49,50,51,52].

Our data suggest that cCMV infection and related symptoms at birth are associated with changes in the immunological signature in the amniotic fluid. Four soluble pro-inflammatory mediators (IP-10, IL-18, ITAC, and TRAIL) and one mediated by EV (IP-10) were increased in case of cCMV infection. Among these proteins, five were related to symptoms at birth (IP-10_internal_, IP-10_surface_, IP-10_soluble_, IL-18 _soluble_, and TRAIL _soluble_). Seven other cytokines, not related to cCMV infection, were significantly associated with symptomatic status at birth; therefore, a pattern for severe infection can be drawn with a specific increase in the presence and concentration of six mediators (IL-18_soluble_, TRAIL_soluble_, CRP_soluble_, TRAIL_surface_, MIG_internal_, and RANTES_internal_).

Most of these relevant cytokines were previously reported in immunodeficient adults with hCMV infection but not within the immunological process involved in cCMV infection based on animal models or in in vitro studies using cell lines cultures [53]. However, the viral host specificity was not considered in these studies, leading to major limitations. TNF-related apoptosis-inducing ligand (TRAIL) induces apoptosis by binding and cross-linking death-domain receptors and activation of caspases [54,55]. In vitro, human CMV-infected fibroblasts co-up-regulated secretion of TRAIL and expression of TRAIL-DR [56]. Independently from CMV, the secretion of TRAIL is inducible by IFN-γ and TNF-α. TRAIL is also secreted by natural killer (NK) cells, an important effector of innate immunity, of which a subset expressing the NKG2C activating receptor is a preferential target for hCMV [57,58,59]. RANTES stands for regulated on activation, normal T cell expressed, and secreted; it is a chemoattractant for T lymphocytes, monocytes, macrophages, and eosinophiles [60]. In vitro, RANTES is released by HCMV-infected fibroblasts [61]. RANTES promotes TNF-α excretion from macrophages, proliferation of NK cells, and T lymphocytes co-activation with MCP-1 [62]. IL-18 is produced by inflammasome, a multimeric protein complex assembled in the cytosol of cells belonging to the innate immune system, especially macrophages [63]. It follows the recognition of pathogen-associated molecular patterns (PAMPs) or damage-associated molecular patterns (DAMPs) [64]. In a murine model, inflammasome activation in the cochlea has been involved in SNHL [65]. IP-10 is a chemokine released by several subset of cells following IFN-γ induction and implicated in regulation of NK cells, monocytes, and lymphocytes [66,67]. IP-10 recruits Th1 cells, which produce IFN-γ, leading to increasing IP-10 concentration. In addition, IP-10 downregulates Th2 cytokine production [68]. MCP-1, monocyte chemoattractant protein-1, is a chemokine involved in attraction and activation of granulocytes, T cells, and monocytes [69]. hCMV infection is associated with high levels of MCP-1 both in vivo and in vitro [61,70,71]. Low levels of MCP-1 seem related to virus survival and chronic infection [72]. Interferon-inducible T-cell alpha chemoattractant (I-TAC) is a chemokine secreted by infected fibroblasts and induced by IFN-γ [73].

Data concerning c reactive protein (CRP) and monokine induced by gamma interferon (MIG) are very limited and restricted to hCMV infection in transplant recipients [74,75]. All relevant cytokines are inducible by type-1 cytokine response, including TNF-α and IFN-γ, suggesting a Th-1 cell polarization of the immune response [68]. Our data contribute to the controversy on the respective importance of Th-1 cell polarization over Th-2 cell polarization involved in viral infection immunity [33,61].

hCMV has developed various mechanisms to evade the immune response, including modulation of cytokine response [28,69,76,77]. The main one uses a CMV homolog of IL-10 to reduce the immune response [76]. CMV also encodes homologs of cytokine receptors, binding RANTES, e.g., which contribute to anti-inflammatory evasion [78]. hCMV reduces the expression of TRAIL-DR, leading to a restriction of the TRAIL/TRAIL-DR pathways and viral proliferation [56]. Figure 4 summarizes the previous discussion, especially interactions between relevant cytokines and immune system cells.

Few studies have investigated prognostic markers associated with cCMV infection in utero. Multi-OMICS approaches offer new perspectives to identify relevant biomarkers. In two studies, proteomic analysis of the amniotic fluid using liquid chromatography–mass spectrometry, LC-MS, or capillary electrophoresis–mass spectrometry, CE-MS, identified a set of potential biomarkers associated with severe fetal infection [79,80]. Unfortunately, there was no recurrent protein between those series. Vorontsov et al. reported a statistical association between severe fetal infection and CRP (fold change: 2.5, *p* = 0.005). We also found an association between CRP-soluble and the severity of fetal infection, suggesting that CRP is a potential prognostic biomarker. One study investigated transcriptomic changes in the case of fetal cCMV infection. Whole transcriptomic analysis using RNA-Seq was performed on 26 samples (13 infected and 13 matched controls) collected between 18 and 23 weeks. Among the 12 most relevant up-regulated genes, there was no recurrent genes with proteins previously identified.

In our study, amniocentesis was performed at a median GA of 22 WG. Recent evidence suggested that an earlier invasive sampling may now be offered to diagnose fetal infection, including amniocentesis from 17 WG and 8 weeks following maternal infection. Our findings should be replicated in a larger prospective cohort, particularly at 17 WG, to be implemented in clinical practice. In addition, a subsequent analysis should be conducted to investigate association between candidate biomarkers and long-term endpoints, including delayed SNHL.

In utero therapy using valaciclovir (8 g/day, 2 g four time a day) was progressively extended from curative (in case of infected fetuses) to preventive treatment (risk of fetal infection following PMI) [81,82]. Our group has recently implemented the diagnosis of fetal infection using CMV-PCR on trophoblast samples obtained by chorionic villus sampling (CVS) at 13–14 weeks [83]. Profiling of inflammatory mediators on infected trophoblast samples could provide additional data on placental immunity and on the pathophysiology of vertical transmission.

The strength of our study is the size of the series and the number of cytokines investigated although the unbalanced ratio of cytokines over cases could be considered a limitation. In our study, cytokines concentrations were assessed using ELISA, the most common technology for measuring proteins concentrations. Most EV cytokines were undetectable or excluded by the pre-processing filtering, possibly because of a lack of sensitivity of ELISA. Considering trafficking of EV, EV cytokines are promising biomarkers, especially for non-invasive tests using maternal blood. New tools based upon single molecule counting technology, such as SiMoA, could be used to detect ultra-low cytokines concentrations to characterize further the profile of EV-cytokines [84].

## 5. Conclusions

Our data suggest that cCMV infection and its severity are associated with differential cytokines expression in amniotic fluid at mid-gestation. These proteins, mainly soluble in amniotic fluid, could be considered as candidate biomarkers of severity in case of fetal infection diagnosed by CMV-PCR.

## Figures and Tables

**Figure 1 viruses-14-02145-f001:**
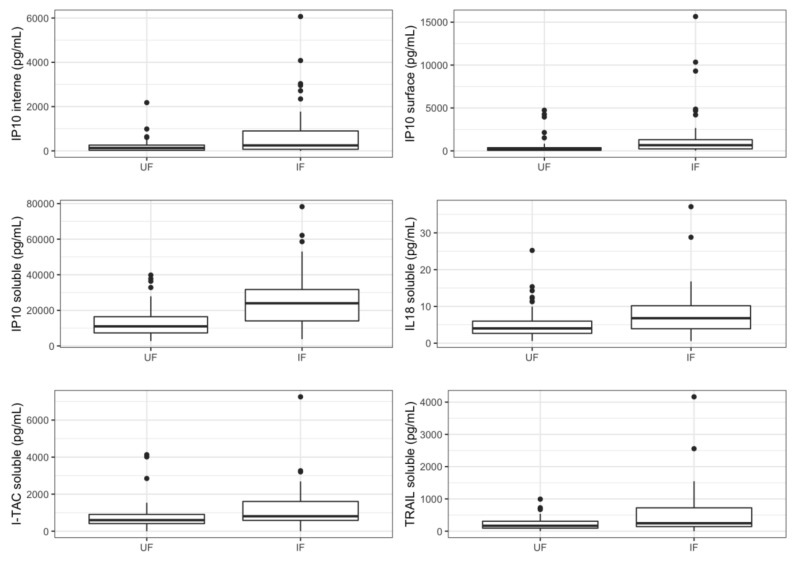
Concentrations of relevant cytokines according to fetal infection. Boxplots represent variations in cytokines’ concentrations (median). UF: uninfected fetuses, IF: infected fetuses.

**Figure 2 viruses-14-02145-f002:**
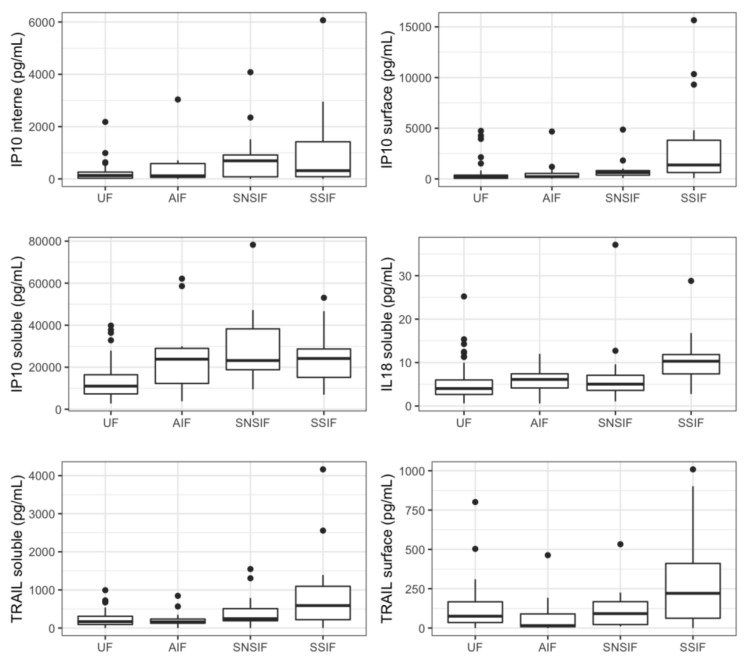
Concentrations of relevant cytokines according to symptomatic state at birth and severity. Boxplots representing represent variations in cytokines’ concentrations (median). UF: unifected fetuses, AIF: asymptomatic infected fetuses, SNSIF: symptomatic and non-severe infected fetuses, SSIF: symptomatic and severe infected fetuses.

**Figure 3 viruses-14-02145-f003:**
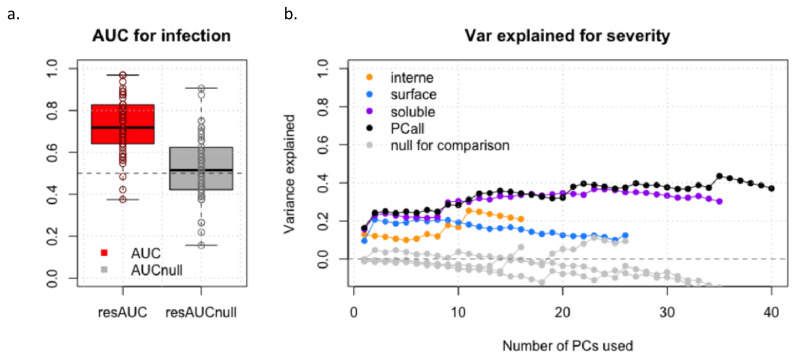
Prediction accuracy of fetal infection (**a**) and of severity (**b**).

**Figure 4 viruses-14-02145-f004:**
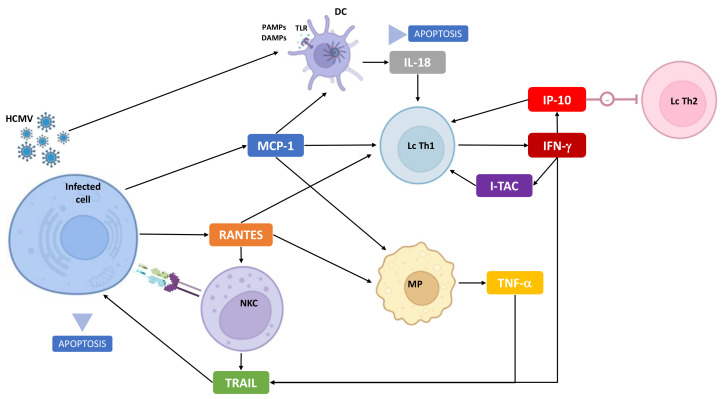
Immunobiology of cCMV: focus on cytokines identified in this study and interactions with infected and immune cells. NKC: natural-killer cell, MP: macrophage, Lc Th1: T-cell involved in Th1 immune response, Lc Th2: T-cell involved in Th2 immune response, DC: dendritic cell, TLR: Toll-like receptor, PAMPS/DAMPs: pathogen-associated molecular patterns/damage-associated molecular patterns.

**Table 1 viruses-14-02145-t001:** Population characteristics.

	Uninfected Fetuses *n* = 40	Infected Fetuses *n* = 40	Asymptomatic Infected Fetuses *n* = 9	Symptomatic Non-Severe Infected Fetuses *n* = 13	Symptomatic and Severe Infected Fetuses *n* = 18	*p*-Value *	*p*-Value **
Maternal age (years)	32 (30; 35)	32 (30; 36)	33 (32; 36)	32 (30; 33)	30 (30; 34)	0.9	0.4
BMI (kg.m^−2^)	20.70 (19.25; 21.75)	21.00 (19.70; 22.70)	20.70 (20.00; 23.40)	21.00 (20.62; 22.00)	21.20 (19.40; 24.10)	0.3	0.9
Nulliparity	7 (19%)	6 (16%)	0 (0%)	3 (23%)	3 (19%)	0.7	0.7
GA amniocentesis (WG)	22.0 (20.0; 25.0)	22.1 (20.6; 24.9)	20.7 (20.3; 22.3)	23.6 (21.9; 27.0)	22.0 (18.8; 25.1)	0.8	0.2

* infected vs. uninfected fetuses; ** symptomatic vs. asymptomatic fetuses.

**Table 2 viruses-14-02145-t002:** Concentrations of relevant cytokines according to fetal infection (median and interquartile).

	Uninfected Fetuses *n* = 40	Infected Fetuses *n* = 40	Asymptomatic Infected Fetuses *n* = 9	Symptomatic Infected Fetuses *n* = 31	Symptomatic Non-Severe Infected Fetuses *n* = 13	Symptomatic and Severe Infected Fetuses *n* = 18	*p*-Value *	*p*-Value **
IP10 interne	135.945 (54.1875; 272.7225)	451.405 (79.335; 956.49)	118.04 (49.58; 605.312)	505.33 (84.27; 1342.97)	717.53 (84.06; 1170.79)	315.52 (84.9775; 1419.735)	0.0037	0.0006
IP10 surface	216.47 (71.6525; 393.6525)	775.625 (300.34; 1680.205)	264.38 (181.94; 704.448)	892.91 (587.435; 2036.44)	740.69 (566.94; 892.91)	1374.01 (629.18; 3813.8825)	0.0138	0.0005
IL18 soluble	4.206 (3.099; 7.05)	7.416 (4.29; 10.698)	6.48 (4.8; 9.24)	8.124 (4.26; 11.562)	5.16 (4.2; 7.2)	10.308 (7.395; 11.85)	0.0234	0.0060
IP10 soluble	10,852.986 (7354.974; 15,979.596)	24,155.274 (13,671.278; 33,171.693)	13,415.432 (7824.636; 28,769.988)	24,195.876 (16,469.694; 36,399.324)	27,967.932 (18,816.576; 39,577.848)	24,155.274 (15,166.275; 28,695.609)	0.0000	0.0001
I-TAC soluble	548.298 (384.462; 831.096)	838.956 (588.57; 1682.418)	822.924 (546.864; 1233.608)	1015.44 (609.666; 1809.216)	1015.44 (628.452; 1651.2)	967.158 (601.602; 1825.788)	0.0418	0.0547
TRAIL soluble	180.4938 (98.5061; 346.6263)	299.2086 (141.0222; 802.9308)	174.888 (141.1776; 224.6172)	415.3464 (155.9106; 888.8652)	245.2572 (140.556; 509.3052)	589.3542 (219.0777; 1095.1194)	0.0123	0.0004

* infected vs. uninfected fetuses; ** symptomatic vs. asymptomatic fetuses.

**Table 3 viruses-14-02145-t003:** Concentrations of relevant cytokines in amniotic fluid according to symptomatic status at birth and severity (median and interquartile).

	Uninfected Fetuses *n* = 40	Infected Fetuses *n* = 40	Asymptomatic Infected Fetuses *n* = 9	Symptomatic Infected Fetuses *n* = 31	Symptomatic Non-Severe Infected Fetuses *n* = 13	Symptomatic and Severe infected Fetuses *n* = 18	*p*-Value *	*p*-Value **
IP10 internal	135.945 (54.1875; 272.7225)	451.405 (79.335; 956.49)	118.04 (49.58; 605.312)	505.33 (84.27; 1342.97)	717.53 (84.06; 1170.79)	315.52 (84.9775; 1419.735)	0.0037	0.0006
MCP1 internal	0 (0; 2.39)	0.375 (0; 5.415)	0 (0; 2.57)	0.75 (0; 5.52)	0.75 (0; 3.66)	2.13 (0; 9.1625)	0.0636	0.0156
MIG internal	0 (0; 9.7825)	0.975 (0; 17.695)	0 (0; 0)	3.7 (0; 18.655)	1.05 (0; 17.49)	10.05 (0; 18.8275)	0.1480	0.0309
RANTES internal	0 (0; 1.8625)	1.65 (0; 5.65)	0 (0; 2.89)	2.08 (0; 6.29)	0 (0; 1.58)	5.505 (1.81; 11.0875)	0.0585	0.0046
IP10 surface	216.47 (71.6525; 393.6525)	775.625 (300.34; 1680.205)	264.38 (181.94; 704.448)	892.91 (587.435; 2036.44)	740.69 (566.94; 892.91)	1374.01 (629.18; 3813.8825)	0.0138	0.0005
CRP surface	2993.57 (1885.88; 5824.6725)	3253.165 (2054.605; 6279.77)	2893.28 (1733.97; 3241.92)	3319.65 (2095.535; 7397.78)	2121.79 (1816.45; 3264.41)	6068.58 (3347.9625; 14,867.88)	0.1388	0.0035
TRAIL surface	77.8392 (36.0268; 216.2042)	90.6635 (21.82125; 218.6225)	25.688 (7.6784; 101.64)	125.625 (27.66375; 229.949)	89.695 (21.015; 133.802)	220.6135 (62.4187; 410.7662)	0.3280	0.0302
IL18 soluble	4.206 (3.099; 7.05)	7.416 (4.29; 10.698)	6.48 (4.8; 9.24)	8.124 (4.26; 11.562)	5.16 (4.2; 7.2)	10.308 (7.395; 11.85)	0.0234	0.0060
IP10 soluble	10,852.986 (7354.974; 15,979.596)	24,155.274 (13,671.278; 33,171.693)	13,415.432 (7824.636; 28,769.988)	24,195.876 (16,469.694; 36,399.324)	27,967.932 (18,816.576; 39,577.848)	24,155.274 (15,166.275; 28,695.609)	0.0000	0.0001
CRP soluble	725.232 (196.254; 1356.666)	757.836 (352.305; 2831.577)	646.2 (356.676; 772.752)	898.992 (374.388; 2997.594)	699.528 (168.756; 898.992)	2841.414 (675.408; 7620.282)	0.0987	0.0044
TRAIL soluble	180.4938 (98.5061; 346.6263)	299.2086 (141.0222; 802.9308)	174.888 (141.1776; 224.6172)	415.3464 (155.9106; 888.8652)	245.2572 (140.556; 509.3052)	589.3542 (219.0777; 1095.1194)	0.0123	0.0004

* infected vs. uninfected fetuses; ** symptomatic vs. asymptomatic fetuses.

## Data Availability

The data presented in this study are available from the corresponding author upon request.

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
