# Peer review of "Cytokine Profiling of Amniotic Fluid from Congenital Cytomegalovirus Infection"

_viruses, 2022, doi:10.3390/v14102145_

Round 1

Reviewer 1 Report

Thank you for asking me to review the manuscript entitled “Profile and concentration of cytokines fractions in amniotic 2 fluid of fetuses infected with cytomegalovirus” by Bourgon et al. The manuscript explores the transplacental milieu and cytokine profile in cCMV. Overall, the biology across the placental interface is poorly understood in most congenital infections, including cCMV and thus this research is welcomed in order to further our understanding of the pathophysiology of cCMV acquisition and disease severity. Overall, the manuscript is well written, methods sound, and informative. Separating results into cytokine profiles based on risk of CMV acquisition in the foetus as well as those predisposing to disease severity makes for practical and simplistic interpretation of results. My main criticism is the lack of discussion on the clinical implication of the authors findings.

Minor Comments:

-in line 61, can the authors be more specific in what they mean by “biologic data”?

-“In such cases and despite changes in definitions over time, around 30% of newborns are considered symptomatic at birth with sensorineural impairment in 10-15%, including hearing loss (SNHL) and vestibulitis and 10-25% suffer more severe neurological damage” These numbers are a bit confusing. Are the 10-15% with sensorineural impairment not included in those with more severe neurological damage or is this 10-25% defined differently?

-“Among biological processes involved in the innate immunity, many cytokines appear essential to control the damage to fetal and adult tissues in cCMV and in immunodeficient adults respectively”

-were all recruited women healthy at baseline or could other conditions confound the cytokine findings?

-formatting of headings on Table 2 need addressing for reader clarity. Also, need to be consistent with language ie. Controls (non-infected)

-what is the relevance of the pre-processing ng filter column in tables 2 and 3?

-”Immune response to cCMV involves both innate and adaptative immunity in the mother, placenta, and fetus at each step of the vertical transmission [32]. Among biological processes involved in the innate immunity, many cytokines appear essential to control cCMV infection, as well as hCMV in immunodeficient adults.” hCMV abbreviation needs defining in the text.

-“Recent data suggested that intraamniotic EV could contain biomarkers relating to bacterial intra-amniotic infection but also congenital diaphragmatic hernia, fetal alcohol spectrum disorder and preterm labor [29,38,43,47–50].” May not be the most relevant of comparisons here.

-“IL-18 is produced by inflammasome, a multimeric protein complex assembled in the cytosol of cells belonging to the innate immune system, especially macrophages,.” (punctuation error)

-MIG needs defining in the text when abbreviation first used

-in appendix B is the term “calcifications of lenticulostriate vessels in basal ganglia” meant to refer to lenticulostriate vasculopathy? This would be a more appropriate description. Although I agree in the interpretation of minor subependymal cysts and/or LSV as asymptomatic, expert opinion does remain somewhat divided.

-were positive OAE findings confirmed by AABR?

-were all abnormal HUS followed up with a brain MRI?

Major Comments:

-I do not have access to the algorithm in the supplementary materials used to determine severity of symptomatic disease which is important in order to interpret results in relation to currently accepted guidelines of asymptomatic v symptomatic and in which symptomatic child would  be offered treatment

-can the authors show data for degree of severity of symptomatic infection to support the following statement: “A pattern with a specific increase in cases of severe symptomatic infection was identified for 6 proteins (IL-18soluble, TRAILsoluble, CRPsoluble, TRAILsurface, MIGinternal and RANTESinternal).” This is a very important finding from a clinical perspective that should be explored.

-It would be nice to have a brief paragraph in the discussion on the clinical application of cytokine responses in cCMV and how this could be used to affect clinical decision making. The authors may want to refer to the article “Valaciclovir to prevent vertical transmission of cytomegalovirus after maternal primary infection during pregnancy: a randomised, double-blind, placebo-controlled trial” by Shahar-Nissan et al. with cytokine responses potentially used as a clinical adjunct in understanding and preventing maternal-foetal transmission of CMV

-do the authors plan to follow these babies as would be interesting if the intrauterine cytokine profile has an impact on later development of SNHL which may not be apparent at birth?

Author Response

Responses to Reviewer 1

First, we thank Reviewer 1 for their support and comments that allowed us to improve the manuscript.

REVIEWER 1, POINT 1

  1. In line 61, can the authors be more specific in what they mean by “biologic data”?
  2. Thank you for your comment.

Fetal thrombocytopenia is the only one prenatal biological data associated with fetal/neonatal prognosis. We have clarified this by adding the specific references.

  1. Page 2, line 61
  2. “biological data, including fetal thrombocytopenia.”

REVIEWER 1, POINT 2

  1. In such cases and despite changes in definitions over time, around 30% of newborns are considered symptomatic at birth with sensorineural impairment in 10-15%, including hearing loss (SNHL) and vestibulitis and 10-25% suffer more severe neurological damage” These numbers are a bit confusing. Are the 10-15% with sensorineural impairment not included in those with more severe neurological damage or is this 10-25% defined differently?
  2. Thank you for your comment.

As suggest, we have clarified this point.

  1. Page 2, line 56-59
  2. “In such cases and despite changes in definitions over time, around 30% of newborns are considered symptomatic at birth with sensorineural impairment (hearing loss (SNHL) and vestibulitis ) in 10-15% and 10-25% suffer more severe neurological damage including intellectual disability or developmental delay [5,9–19].”

REVIEWER 1, POINT 3

  1. Among biological processes involved in the innate immunity, many cytokines appear essential to control the damage to fetal and adult tissues in cCMV and in immunodeficient adults respectively.
  2. Thank you for your comment, we supposed that this sentence was not understandable. We have clarified this point.
  3. Page 2, line 66-67
  4. “Among biological processes involved in the innate immunity, many cytokines are in-volved in immune control of cCMV infection in fetuses and immunodeficient adults [28].”

REVIEWER 1, POINT 4

  1. Were all recruited women healthy at baseline or could other conditions confound the cytokine findings?
  2. Thank you for your comment. Indeed, we checked that all patients did not have pas medical history, including immune disorders...
  3. Page 2, line 83-83
  4. “All women have no relevant medical history, especially no immune disorders or treatment affecting immunity.”

REVIEWER 1, POINT 5

  1. Formatting of headings on Table 2 need addressing for reader clarity. Also, need to be consistent with language ie. Controls (non-infected)
  2. Thank you for your comment.

We performed the suggested queries.

  1. Pages 4 and 5.
  2. Tables 1, 2 and 3.

REVIEWER 1, POINT 6

  1. What is the relevance of the pre-processing ng filter column in tables 2 and 3?
  2. Thank you for your comment.

We performed the suggested queries to clarify our tables.

  1. Pages 4 and 5.
  2. Tables 2 and 3.

REVIEWER 1, POINT 6

  1. Immune response to cCMV involves both innate and adaptative immunity in the mother, placenta, and fetus at each step of the vertical transmission [32]. Among biological processes involved in the innate immunity, many cytokines appear essential to control cCMV infection, as well as hCMV in immunodeficient adults.” hCMV abbreviation needs defining in the text.
  2. Thank you for your comment.
  3. Page 7, lines 237-239
  4. “Among biological processes involved in the innate immunity, many cytokines appear essential to control cCMV infection, as well as human CMV (hCMV) in immunodeficient adults.”

REVIEWER 1, POINT 7

  1. Recent data suggested that intraamniotic EV could contain biomarkers relating to bacterial intra-amniotic infection but also congenital diaphragmatic hernia, fetal alcohol spectrum disorder and preterm labor [29,38,43,47–50].” May not be the most relevant of comparisons here.
  2. Thank you for your comment.

We performed the suggested queries.

  1. Page 7, lines 247-249
  2. “Recent data suggested that intraamniotic EV could contain biomarkers relating to a wide spectrum of fetal disorders including bacterial intra-amniotic infection [30,40,45,49–52].”

REVIEWER 1, POINT 8

  1. IL-18 is produced by inflammasome, a multimeric protein complex assembled in the cytosol of cells belonging to the innate immune system, especially macrophages,.” (punctuation error)
  2. Thank you for your comment.

We performed the suggested queries.

  1. Page 8, lines 273-275
  2. “IL-18 is produced by inflammasome, a multimeric protein complex assembled in the cytosol of cells belonging to the innate immune system, especially macrophages [63].”

REVIEWER 1, POINT 9

  1. MIG needs defining in the text when abbreviation first used.
  2. Thank you for your comment.
  3. Page 8, lines 287-289
  4. “Data concerning c reactive protein (CRP) and monokine induced by gamma inter-feron (MIG) are very limited and restricted to hCMV infection in transplant recipients [74,75].”

REVIEWER 1, POINT 10

  1. in appendix B is the term “calcifications of lenticulostriate vessels in basal ganglia” meant to refer to lenticulostriate vasculopathy? This would be a more appropriate description. Although I agree in the interpretation of minor subependymal cysts and/or LSV as asymptomatic, expert opinion does remain somewhat divided.
  2. Thank you for your comment.

We are working on a publication on the prognosis of fetuses with these signs. We can refer to it in a future response if necessary.

  1. Page 12, line 1044
  2. “…subependymal cysts and/or calcifications of lenticulostriate vasculopathy”

REVIEWER 1, POINT 11

  1. were positive OAE findings confirmed by AABR? were all abnormal HUS followed up with a brain MRI?
  2. Thank you for your questions.

Indeed, abnormal OAE were confirmed in all cases by AABR. Postnatal brain MRI was offered in case of abnormal TUS and abnormal clinical examination.

  1. Page 12, lines 1042-1043
  2. “…hearing tests (otoacoustic emissions (OAE) and/or automated auditory brainstem response (AABR) in case of abnormal OAE), fundoscopic examination (FE) and neonatal transcranial US examination (TUS). In case of abnormal TUS and clinical examination a brain MRI was offered.”

REVIEWER 1, POINT 12

  1. I do not have access to the algorithm in the supplementary materials used to determine severity of symptomatic disease which is important in order to interpret results in relation to currently accepted guidelines of asymptomatic v symptomatic and in which symptomatic child would be offered treatment.
  2. Thank you for your comment.

We have added in appendix B the algorithm to define symptomatic cases and severe symptomatic cases. Unfortunately, in this study, all severe cases were died (TOP or IUFD) and severity confirmed by postmortem examination.

  1. Page 12, lines 1040-1050
  2. “Neonates were asymptomatic if there is no growth restriction (z-score <1.28 using the Inter-growth standards), no abnormal clinical features, no biological abnormalities (thrombocytope-nia, hepatic cytolysis, or mixed hyperbilirubinemia), no abnormality on OAE/AABR, FE or TUS. Fetuses with isolated unilateral minor cerebral features (subependymal cysts and/or calcifica-tions of lenticulostriate vasculopathy) were considered asymptomatic. The others were consid-ered symptomatic. Fetuses harboring at least one severe cerebral feature (cortical abnormalities, ventriculomegaly > 15 mm, enlarged percicerebral spaces or microcephaly) independently of the pregnancy outcome (termination of pregnancy, intrauterine fetal demise, perinatal death, or live birth) were considered as severe symptomatic fetuses. In case of termination of pregnancy, intra-uterine fetal demise or perinatal death, a postmortem examination was systematically offered. In all lethal cases, postmortem examination confirmed a severe infection with focal necrosis.”

REVIEWER 1, POINT 13

  1. can the authors show data for degree of severity of symptomatic infection to support the following statement: “A pattern with a specific increase in cases of severe symptomatic infection was identified for 6 proteins (IL-18soluble, TRAILsoluble, CRPsoluble, TRAILsurface, MIGinternal and RANTESinternal).” This is a very important finding from a clinical perspective that should be explored.
  2. Thank you for your question.

The specific increase of this set of proteins is now clearly shows in Figure 2 and Table 3.

  1. Page 6 and 7.

REVIEWER 1, POINT 14

  1. It would be nice to have a brief paragraph in the discussion on the clinical application of cytokine responses in cCMV and how this could be used to affect clinical decision making. The authors may want to refer to the article “Valaciclovir to prevent vertical transmission of cytomegalovirus after maternal primary infection during pregnancy: a randomised, double-blind, placebo-controlled trial” by Shahar-Nissan et al. with cytokine responses potentially used as a clinical adjunct in understanding and preventing maternal-foetal transmission of CMV
  2. Thank you for your comment.

We have modified our discussion to include a part about clinical implication at a time of preventive in utero therapy.

  1. Page 9-10, lines 891-941,
  2. “Few studies have investigated prognostic markers associated with cCMV infection in utero. Multi-OMICS approaches offer new perspectives to identify relevant biomarkers. In two studies, proteomic analysis of the amniotic fluid using liquid chromatography–mass spectrometry, LC-MS, or capillary electrophoresis–mass spectrometry, CE-MS, identified a set of potential biomarkers associated with severe fetal infection [79,80]. Unfortunately, there was no recurrent protein between those series. Vorontsov and al. reported a sta-tistical association between severe fetal infection and CRP (fold change:2.5, p=0.005). We also found an association between CRP-soluble and the severity of fetal infection; sug-gesting that CRP is a potential prognostic biomarker. One study investigated tran-scriptomic changes in case of fetal cCMV infection. Whole transcriptomic analysis using RNA-Seq was performed on 26 samples (13 infected and 13 matched controls) collected between 18 and 23 weeks. Among the 12 most relevant up-regulated genes, there was no recurrent genes with proteins previously identified.

In our study, amniocentesis was performed at a median GA of 22 WG. Recent ev-idence suggested an earlier invasive sampling may now be offered to diagnose fetal infection, including amniocentesis from 17 WG and 8 weeks following maternal infection. Our findings should be replicated in a larger prospective cohort, particularly at 17 WG, to be implemented in clinical practice. In addition, a subsequent analysis should be con-ducted to investigate association between candidate biomarkers and long-term endpoints, including delayed SNHL.

In utero therapy using valaciclovir (8 g/day, 2g four time a day) was progressively extended from curative (in case of infected fetuses) to preventive treatment (risk of fetal infection following PMI) [81,82]. Our group has recently implemented the diagnosis of fetal infection using CMV-PCR on trophoblast samples obtained by chorionic villus sampling (CVS) at 13-14 weeks’ [83]. Profiling of inflammatory mediators on infected trophoblast samples could provide additional data on placental immunity and on the pathophysiology of vertical transmission.”

REVIEWER 1, POINT 15

  1. Do the authors plan to follow these babies as would be interesting if the intrauterine cytokine profile has an impact on later development of SNHL which may not be apparent at birth?
  2. Thank you for your comment.

We plan to replicate our analyses for long-term outcomes.

  1. Page, lines 933-934
  2. In addition, a subsequent analysis should be conducted to investigate association between candidate biomarkers and long-term endpoints, including delayed SNHL."

Reviewer 2 Report

This is an interesting manuscript that contributes to the field by evaluating cytokine concentrations in amniotic fluid of children with congenital CMV (cCMV) infection.  The authors examined cytokines in the soluble fraction of the amniotic fluid and also isolated exosomes (EV, extracellular vesicles), which were evaluated for either surface cytokines or solubilized for measurement of internal cytokines.  The results show that a few cytokines correlated with cCMV.  IP-10 was notable because it was elevated in the soluble fraction, in EVs, and on the surface of EVs.  Overall, the manuscript provides an exciting glimpse into biomarkers that could help identify those children at risk of severe outcomes due to cCMV infection.  However, attention to several items is necessary before the manuscript could be considered acceptable for publication:

 English language editing is required.  There are many places where verb tense is incorrect and where plurals are used incorrectly, starting with the title:  “Profile and concentration of cytokine fractions in amniotic fluid of fetuses infected with cytomegalovirus”  This reviewer suggests that the tile be reconsidered, perhaps “Cytokine profiling of amniotic fluid from congenital  cytomegalovirus infection”.

2.       Section 2.5 of the methods also needs editing – it should be Cytokine concentration measurement (line 120), antibody (line 123), in 96-well flat bottom plates (remove a, line 127).  Line 136 is 5P five parameter?  Write out

3.       Table formatting – the tables are extremely difficult to read, the values for each column span several lines.  Consider using landscape format or reducing text size to fit better.  Also, the description of p-vale should be a footnote, not spread over 5 lines in top of table (Table 1, 2)

4.       Figure 1 – indicate units on graphs (pg/ml?)

5.       Figure 2  - spelling – it should be symptoms, also, it would be easier for the reader if there was a legend for what 1, 2, 3, 4 are rather than just written out in the figure caption.

6.       The discussion is poorly referenced and missing many critical citations.  This is unacceptable.  There are more than a dozen references about EVs and almost no references at all about the claims the authors make about findings during CMV infection that support their observations.  Notably, lines 260-262 have no citation, nor is there any reference provided for details about the NKG2C receptors as a preferential target of CMV.  One review referenced in line 257 is insufficient.   Where are the references for high levels of MCP-1 during CMV infection or for cmvIL-10?  The receptor for RANTES, which this reviewer assumes to be US28 (line 289-290) , is not mentioned in the paper and the no reference is provided.  

7.       Figure 4 is confusing.  What type of cell is infected?  More explanation for the diagrams is needed.  It is not clear what the authors are trying to depict, nor is it clear how this figure summarizes the discussion as noted in line 292.   

Author Response

Responses to Reviewer 2

First, we thank Reviewer 2 for their support and comments that allowed us to improve the manuscript.

REVIEWER 2, POINT 1

  1. English language editing is required
  2. Thank you for your comment.

English have been reviewed by co-authors, American native.

REVIEWER 2, POINT 2

  1. There are many places where verb tense is incorrect and where plurals are used incorrectly, starting with the title:  “Profile and concentration of cytokine fractions in amniotic fluid of fetuses infected with cytomegalovirus”  This reviewer suggests that the tile be reconsidered, perhaps “Cytokine profiling of amniotic fluid from congenital  cytomegalovirus infection”.
  2. Thank you for your comment.

We performed the suggested queries.

  1. Page 1, line 2
  2. “Cytokine profiling of amniotic fluid from congenital cytomegalovirus infection”

REVIEWER 2, POINT 2

  1. Section 2.5 of the methods also needs editing – it should be Cytokine concentration measurement (line 120), antibody (line 123), in 96-well flat bottom plates (remove a, line 127).  Line 136 is 5P five parameter? 
  2. Thank you for your comment.

We performed the suggested queries.

  1. Page 2, lines 157-175
  2. “2.5. Cytokines concentrations measurement

Inflammatory mediator concentrations were determined using an in-house multi-plexed bead-based assay as previously described, with minor modifications, for 38 cy-tokines (list available in Supplementary method) [30,32]. All antibody pairs and protein standards were purchased from R&D Systems (Minneapolis MN, USA), except those for IFN-β (Abcam, Cambridge, UK). Magnetic beads (Luminex Corporation, Austin, TX) with distinct spectral signatures (regions) were coupled to analyte-specific capture anti-bodies according to manufacturer’s recommendations in 96-well flat bottom plates (Nunc, ThermoFisher) and incubated at 4°C overnight. Plates were washed on a magnetic plate washer (405 TS, Biotek Winooski, VT, USA) followed by incubation with polyclonal bi-otinylated anti-analyte antibodies and streptavidin-phycoerythrin (ThermoFisher Sci-entific). Beads were resuspended in PBS and read on a Luminex 200 analyzer (Luminex Corporation) with acquisition of 100 beads for each region and analyzed using Bioplex Manager software (Bio-Rad Laboratories, Hercules, CA).

Analyte concentrations (pg/ml) were determined using five parameters regression algorithms and expressed as the mean pg/ml ± S.E. Concentrations were corrected for dilution by ExoQuick-TC™ or Triton™ X-100. Extracellular vesicle luminal content (‘internal fraction’) was calculated as [analyte content of lysed vesicle] - [analyte content of intact vesicles (‘surface fraction’)].”

REVIEWER 2, POINT 3

  1. Table formatting – the tables are extremely difficult to read, the values for each column span several lines.  Consider using landscape format or reducing text size to fit better.  Also, the description of p-vale should be a footnote, not spread over 5 lines in top of table (Table 1, 2)
  2. Thank you for your comment.

We performed the suggested queries.

If landscape format is possible for the Tables and Figure 2, we will used it with the Editor.

  1. Pages 4, 5 and 6, Tables 1, 2 and 3

REVIEWER 2, POINT 3

  1. Figure 1 – indicate units on graphs (pg/ml?)
  2. Thank you for your comment.

We performed the suggested queries.

  1. Figures 1 and 2

REVIEWER 2, POINT 4

  1. Figure 2  - spelling – it should be symptoms, also, it would be easier for the reader if there was a legend for what 1, 2, 3, 4 are rather than just written out in the figure caption.
  2. Thank you for your comment.

We performed the suggested queries.

  1. Figures 1 and 2

REVIEWER 2, POINT 5

  1. The discussion is poorly referenced and missing many critical citations.  This is unacceptable.  There are more than a dozen references about EVs and almost no references at all about the claims the authors make about findings during CMV infection that support their observations.  Notably, lines 260-262 have no citation, nor is there any reference provided for details about the NKG2C receptors as a preferential target of CMV.  One review referenced in line 257 is insufficient.   Where are the references for high levels of MCP-1 during CMV infection or for cmvIL-10?  The receptor for RANTES, which this reviewer assumes to be US28 (line 289-290), is not mentioned in the paper and the no reference is provided.  
  2. Thank you for your comment.

We have corrected this major error due to the style changes.

We have added all the references we collected during our literature review. 

We hope we have met your expectations.

  1. Pages 8-10

REVIEWER 2, POINT 6

  1. Figure 4 is confusing.  What type of cell is infected?  More explanation for the diagrams is needed.  It is not clear what the authors are trying to depict, nor is it clear how this figure summarizes the discussion as noted in line 292.
  2. Thank you for your question.

This is a summarized illustration of the published data about cytokines and immune cells in hCMV infection.

We suggest, it is too early to identify a particular cell type, especially in case of cCMV infection. Future investigations should clarify whether these relations are applicable to all infected cell types.

  1. Figure 4
